# OpenML Benchmarking Suites

**Bernd Bischl**[1]*, **Giuseppe Casalicchio**[1], **Matthias Feurer**[2], **Pieter Gijsbers**[3], **Frank Hutter**[2,4],
**Michel Lang**[5], **Rafael G. Mantovani**[6], **Jan N. van Rijn**[7], **Joaquin Vanschoren**[3]

[1] Department of Statistics, LMU Munich, Germany
[2] Department of Computer Science, University of Freiburg, Germany
[3] Department of Computer Science, Eindhoven University of Technology, the Netherlands
[4] Bosch Center for Artificial Intelligence
[5] Department of Statistics, TU Dortmund University, Germany
[6] Federal Technology University Paraná (UTFPR), Brazil
[7] Leiden Institute of Advanced Computer Science (LIACS), Leiden University, the Netherlands

## Abstract

Machine learning research depends on objectively interpretable, comparable, and reproducible algorithm benchmarks. We advocate the use of curated, comprehensive *suites* of machine learning tasks to standardize the setup, execution, and reporting of benchmarks. We enable this through software tools that help to create and leverage these benchmarking suites. These are seamlessly integrated into the OpenML platform, and accessible through interfaces in Python, Java, and R. OpenML benchmarking suites (a) are easy to use through standardized data formats, APIs, and client libraries; (b) come with extensive meta-information on the included datasets; and (c) allow benchmarks to be shared and reused in future studies. We then present a first, carefully curated and practical benchmarking suite for classification: the **OpenML C**urated **C**lassification benchmarking suite 20**18** (OpenML-CC18). Finally, we discuss use cases and applications which demonstrate the usefulness of OpenML benchmarking suites and the OpenML-CC18 in particular.

## 1   Introduction

Algorithm benchmarks shine a beacon for machine learning research. They allow us, as a community, to track progress over time, identify challenging issues, to raise the bar and learn how to do better. To learn as much as possible from them, they must include well-designed, challenging sets of tasks, be easily accessible and practical to use. Evaluations of algorithms on these tasks should be performed in standardized ways to support a rigorous analysis and clear conclusions. And above all, these evaluations must be easy to find, easily interpretable, reproducible, and directly comparable to evaluations run by other scientists.

The OpenML platform [Vanschoren et al., 2013] already serves thousands of datasets together with tasks in a machine-readable way. Tasks define the evaluation procedure for a specific dataset. Concretely, a task contains a reference to a dataset, information on the task type (e.g., classification or regression), the target feature (in the case of supervised problems), the evaluation procedure (e.g., k-fold CV, hold-out), the specific splits for that procedure, and the target performance metric, which together allow for reproducible evaluation schemes. OpenML is also integrated into many machine learning libraries, so that fine details about machine learning models (or pipelines) and their performance evaluations can be automatically collected. This integration allows experiments to be automatically shared and organized on the platform, linked to the underlying datasets and tasks.

---

*Authors are ordered alphabetically. Correspondence to {bernd.bischl | giuseppe.casalicchio}@lmu.de.

However, OpenML did not yet facilitate the simple creation and sharing of well-designed benchmark suites and results of experiments ran on them.

We introduce a novel benchmarking layer on top of OpenML, fully integrated into the platform and its APIs, that streamlines the creation of *benchmarking suites*, i.e., collections of tasks designed to thoroughly evaluate algorithms. These suites can then be easily imported, used in systematic benchmarking experiments, and the results can be automatically shared and organized on the OpenML platform, where they can be easily searched, reused and compared to the results of others. We develop tools that allow for creating a well-defined benchmark suite, and propose a new benchmark suite designed with these tools: the **C**urated **C**lassification benchmarking suite 20**18** (OpenML-CC18).

In short, the contributions of this paper are as follows: (1) we advocate the use of curated, comprehensive *suites* of machine learning tasks (i.e., a dataset with meta-information about the evaluation procedure) to standardize benchmarking, (2) we provide software tools to easily create and use these benchmarking suites, (3) we propose a new benchmark suite (OpenML-CC18), (4) have a closer look at an existing AutoML benchmark suite, and (5) discuss their impact on machine learning research. [1]

We will first discuss related work. Next, we explain how OpenML benchmarking suites work and how to use them in practice. We then present the OpenML-CC18 and review other benchmarking suites, including the AutoML benchmark. Finally, we discuss the impact of benchmarking suites on machine learning research and present our conclusions.

## 2 A Brief History of Benchmarking Suites

The machine learning field has long recognized the importance of dataset repositories. The UCI repository [Dheeru and Taniskidou, 2017] and LIBSVM [Chang and Lin, 2011] offer a wide range of datasets. Many more focused repositories also exist, such as UCR [Chen et al., 2015] for time series data and Mulan [Tsoumakas et al., 2011] for multilabel datasets. Some repositories also provide programmatic access. Kaggle.com and PMLB [Olson et al., 2017] offer a Python API for downloading datasets, skdata [Bergstra et al., 2015] offers a Python API for downloading computer vision and natural language processing datasets, and KEEL [Alcala et al., 2010] offers a Java and R API for imbalanced classification and datasets with missing values.

Several platforms can also link datasets to reproducible experiments (similar to OpenML tasks). Reinforcement learning environments such as the OpenAI Gym [Brockman et al., 2016] run and evaluate reinforcement learning experiments, the COCO suite standardizes benchmarking for black-box optimization [Hansen et al., 2020] and ASLib provides a benchmarking protocol for algorithm selection [Bischl et al., 2016a]. The Ludwig Benchmarking Toolkit orchestrates the use of datasets, tasks and models for personalized benchmarking and so far integrates the Ludwig deep learning toolbox [Narayan et al., 2021]. PapersWithCode maintains a manually updated overview of model evaluations linked to datasets.

Although for many years machine learning researchers have benchmarked their algorithms on some subset of these datasets, this has not yet led to standardized benchmarks that can be easily compared between individual studies. This often results in suboptimal shortcuts in study design, producing rather small-scale experiments that should be interpreted with caution [Aha, 1992], are hard to reproduce [Pedersen, 2008, Hutson, 2018], and even lead to contradictory results [Keogh and Kasetty, 2003]. An often criticized aspect is the competitive mindset in benchmarking which focuses too much on dominating the state-of-art on a few datasets, instead of a rigorous and informative analysis of large-scale studies, including negative results where popular algorithms fail [Sculley et al., 2018].

## 3 OpenML

OpenML is a collaborative platform that allows anyone to share new datasets, and enables anyone to easily import these datasets and subsequently share their own models and experiments run on them. It organizes everything based on four fundamental, machine-readable building blocks: (1) the *data*, (2) the machine learning *task* to be solved, specifying the dataset, the task type (e.g., classification or

---

[1]We previously published a preprint on arXiv, which has already been used in new research. This is the reason we can both introduce OpenML-CC18 and benchmark suites technology, but also review their use. For example, the AutoML benchmark suite was created with the technology described in this paper (and the preprint).

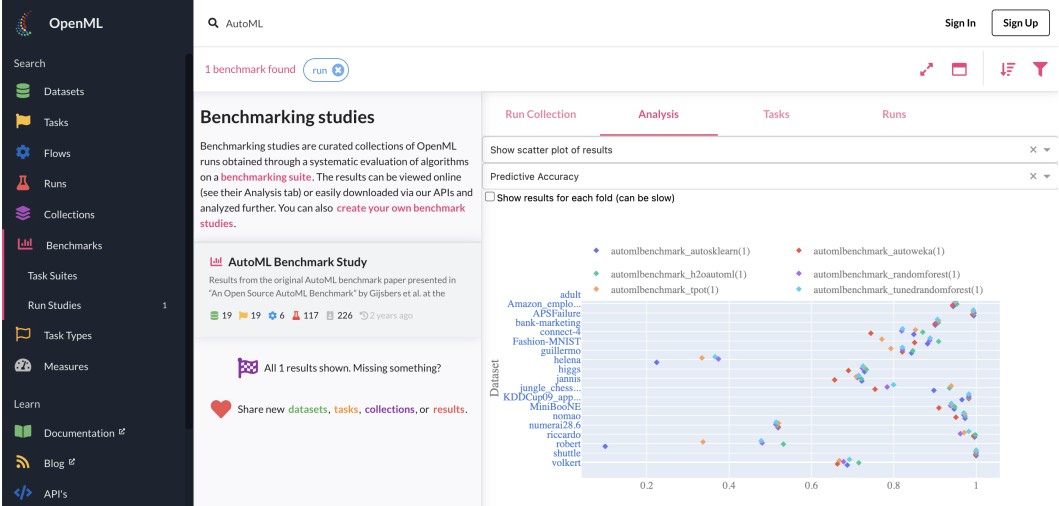

Figure 1: OpenML website showing a list of benchmark studies on the left, and interactive exploration of the results of the AutoML Benchmark (see Section 7.1) on the right. Can be viewed online at `https://www.openml.org/s/226`.

regression), the target feature (in the case of supervised problems), the evaluation procedure (e.g., k-fold CV, hold-out), the specific splits for that procedure, and the target performance metric (3) the *flow* which specifies a machine learning pipeline that solves the *task*, and (4) the *run* that contains experiment results (e.g., predictions and performance evaluations) when a *flow* is executed on a *task* (see Vanschoren et al. [2013] for more details). OpenML goes beyond the platforms mentioned in Section 2, as it includes extensive programmatic access to all datasets, tasks, flows, and runs, comprehensive logging of experiments, and automated sharing of results, which have enabled the collection of millions of publicly shared and reproducible experiments, linked to the exact datasets, machine learning pipelines and hyperparameter settings. OpenML offers bindings with the Java, Python and R ecosystems [van Rijn, 2016, Feurer et al., 2021b, Casalicchio et al., 2017] to provide easy integration in common machine learning tools, workflows, and environments. An introduction and detailed information can be found on `https://docs.openml.org`.

## 4 OpenML Benchmarking Suites

As with any platform where people can upload new datasets, an overwhelming amount and variety of datasets is available, and it can be unclear how well they are curated. We designed OpenML benchmarking suites as a remedy to allow researchers to compile and publish well-defined collections of curated tasks and datasets, and collect benchmarking results from many scientists in a single place. More precisely, we define:

*An OpenML benchmarking suite is a set of OpenML tasks carefully selected to evaluate algorithms under a precise set of conditions.*

Using a set of *tasks* instead of a set of *datasets* makes experiments performed on them comparable and reproducible. Compared to other (static) collections of datasets, the use of OpenML benchmarking suites has the following advantages:

- Easy creation of benchmarks (see Section 5.1): OpenML hosts thousands of datasets, and scientists can easily filter them down to those needed for their benchmarks (see Sections 6 and 7 for examples).
- Convenient access and sharing of suites: Each suite receives a unique ID, which can be used to retrieve the suite via APIs, and via its own webpage. Figure 1 illustrates how results collected on these suites can be explored online.
- Permanence and provenance: Because benchmarking suites are its own entity on OpenML, it is clear who created them (provenance). It also guarantees no one but the original creator can edit or remove the suite (permanence), this is an advantage over the previously used community tagging mechanism which allowed any user to add tasks to a suite.

- Community of practice: Curated benchmark suites allow scientists to thoroughly benchmark their machine learning methods without having to worry about finding and selecting datasets for their benchmarks.
- Building on existing suites: Scientists can extend, subset, or adapt existing benchmarking suites to correct issues, raise the bar, or run personalized benchmarks.
- Reproducibility of benchmarks: Based on machine-readable OpenML *tasks*, with detailed instructions for evaluation procedures and train-test splits, shared results are comparable and reproducible.
- Conducting benchmark studies: After creating an OpenML benchmarking suite, existing and new experiments (*runs*) on the underlying *tasks* can be associated with the suite. This is also illustrated in Figure 3. Such data reuse bootstraps the creation of new benchmark studies that can analyze existing machine learning algorithms in new ways, or to design new challenging benchmark suites.
- Collaborative work: OpenML benchmarking suites benefit from the OpenML community, where users can help to identify and report bugs and errors in the contained datasets.
- Dynamic benchmarks: Benchmarks are never perfect, and when used for a long time, scientists may overfit on specific sets of tasks. However, benchmarking suites can be easily corrected and extended over time (e.g., on a yearly basis), leading to dynamic benchmarks that respond to novel concerns, and evaluate methods on new and ever more challenging tasks. More than providing a snapshot, this allows longitudinal studies that truly track progress over time.

# 5    How to Use OpenML Benchmarking Suites

To realize all these benefits, we have developed a series of extensions to the OpenML platform:[2]

- We added the concepts of a 'benchmark suite' as a collection of *tasks*, and a 'benchmark study' as a collection of benchmark results (*runs*) obtained on them.
- We added data filtering procedures to the APIs and website that allow researchers to exactly specify the constraints for tasks to be included in a benchmark suite.
- We provide scripts and notebooks that facilitate the creation and quality assessment of benchmark suites. For instance, they filter out datasets that are modeled too easily, and hence cannot be used to differentiate between most algorithms (see Section 5.1).
- Certain types of datasets, such as multilabel, time series, or artificial datasets, may require additional care. We added collaborative and automated annotation (tagging) to filter such datasets accordingly.

In the following, we discuss the three main use cases for benchmarking suites, i.e., creating new suites, retrieving existing suites, and running benchmarks. We provide code examples on how to retrieve, iterate the contents of a benchmark suite and run machine learning algorithms on it in Figure 2.[3]

## 5.1    Creating New Suites

To collect data sets for a new suite, one usually starts by determining a list of constraints that datasets or tasks should adhere to (e.g., have a minimal size, a limited amount of class imbalance, and not be a time series). This is often an iterative refinement process, during which the distribution of currently selected tasks can be visualized, and any existing benchmarking results on these tasks can be retrieved. An example of this workflow is illustrated in the provided notebook.[4] The final selection of tasks can then be used to create a new benchmark suite. Each benchmark suite is assigned a unique id and an overview webpage with a description and an analysis dashboard (e.g., `https://www.openml.org/s/99`). The description text can be used to describe the goals and design criteria, provide links to external resources, and address any ethical concerns that should be taken into consideration when using the benchmark suite. We give an exemplary curation protocol in the Appendix.

## 5.2    Retrieving Existing Suites

Existing benchmark suites can be easily downloaded via any of the OpenML client libraries using its unique id or alias (see Figure 2). The tasks and datasets are all uniformly formatted, and come

---

[2]All code is open, BSD-3 licenced, and available on `https://github.com/openml`

[3]More detailed and up-to-date instructions can be found on: `https://docs.openml.org/benchmark`

[4]Notebooks can be found at `https://github.com/openml/benchmark-suites`

with extensive meta-data to streamline the execution of benchmarks on them. For instance, if a dataset contains missing values, this is indicated in a machine-readable way so that researchers

```python
from openml import config, study, tasks, runs, extensions
from sklearn import compose, impute, metrics, pipeline, preprocessing, tree

clf = pipeline.make_pipeline(
    compose.make_column_transformer(
        (impute.SimpleImputer(), extensions.sklearn.cont),
        (preprocessing.OneHotEncoder(handle_unknown='ignore'), extensions.sklearn.cat),
    ),
    tree.DecisionTreeClassifier(max_depth=1)
) # build a fast and simple classification pipeline

benchmark_suite = study.get_suite('OpenML-CC18') # obtain the benchmark suite
# config.apikey = 'FILL_IN_OPENML_API_KEY'       # uploading to OpenML requires an API key

run_ids = []
for task_id in benchmark_suite.tasks:            # iterate over all tasks
    task = tasks.get_task(task_id)               # download the OpenML task
    X, y = task.get_X_and_y()                    # get the data (not used in this example)
    run = runs.run_model_on_task(clf, task)      # run classifier on splits given by the task
    score = run.get_metric_fn(metrics.accuracy_score) # compute and print the accuracy score
    print(f'Data set: {task.get_dataset().name}; Accuracy: {score.mean():.2}')
    run.publish()
    run_ids.append(run.id)

benchmark_study = study.create_study(            # create a study to share the set of results
    name="CC18-Example",
    description="An example study reporting results of a decision stump.",
    run_ids=run_ids,
    benchmark_suite=benchmark_suite.id
)
benchmark_study.publish()
print(f"Results are stored at {benchmark_study.openml_url}")
```

(a) Python, available as pypi package OpenML

```java
public static void runTasksAndUpload() throws Exception {
  OpenmlConnector openml = new OpenmlConnector("FILL_IN_OPENML_API_KEY");
  Study benchmarksuite = openml.studyGet("OpenML-CC18", "tasks");   // obtain the benchmark suite
  Classifier tree = new REPTree();                                  // build a Weka classifier
  for (Integer taskId : benchmarksuite.getTasks()) {               // iterate over all tasks
    Task t = openml.taskGet(taskId);                                // download the OpenML task
    Instances d = InstancesHelper.getDatasetFromTask(openml, t);   // obtain the dataset
    Pair<Integer, Run> result = RunOpenmlJob.executeTask(openml, new WekaConfig(), taskId, tree);
    Run run = openml.runGet(result.getLeft());
  }
}
```

(b) Java, available on Maven Central with artifact id org.openml.openmlweka

```r
library(OpenML)                                  # requires at least package version 1.8
library(mlr)
lrn = makeLearner('classif.rpart')              # construct a simple CART classifier
bsuite = getOMLStudy('OpenML-CC18')             # obtain the benchmark suite
task.ids = extractOMLStudyIds(bsuite, 'task.id') # obtain the list of suggested tasks
for (task.id in task.ids) {                      # iterate over all tasks
  task = getOMLTask(task.id)                     # download single OML task
  data = as.data.frame(task)                     # obtain raw data set
  run = runTaskMlr(task, learner = lrn)          # run constructed learner
  setOMLConfig(apikey = 'FILL_IN_OPENML_API_KEY')
  upload = uploadOMLRun(run)                      # upload and tag the run
}
```

(c) R, available on CRAN via package OpenML

Figure 2: Complete code examples, in different programming languages, of how any benchmarking suite (here the 'OpenML-CC18' suite) can be downloaded and used to evaluate a given algorithm. The Python code also creates a new benchmark study and shares all results. Uploading requires a (free) API key.

can automatically adjust for this when running their algorithms. Datasets can be investigated using exploratory data analysis tools, and existing runs on these tasks can be downloaded and analyzed.

## 5.3 Running Benchmarks

After retrieving the tasks from a suite, new experiments can be conducted locally. As illustrated in Figure 2, this is easiest with the readily integrated machine learning libraries, such as scikit-learn [Pedregosa et al., 2011], mlr [Bischl et al., 2016b] or its successor mlr3 [Lang et al., 2019], and Weka [Hall et al., 2009]. Integrations for deep learning libraries are under development, and we welcome further open source integrations.[5] Custom code can often be wrapped, e.g., using the scikit-learn interface.

The results of these experiments (runs) can also (optionally) be bundled in a benchmark study and published on OpenML, as illustrated for Python in Figure 2. Runs include all experiment details, including hyperparameter configurations, in a structured way. This allows entire communities of scientists to bring together benchmarks of a wide range of algorithms, all evaluated uniformly on the same tasks, in a single place where they can be directly compared on predictive performance and analysed in novel ways. Figure 3 visualizes the results of 3.8 million runs collected on a single benchmarking suite, which we will discuss next.

# 6 OpenML-CC18

To demonstrate the functionality of OpenML benchmarking suites, we created a first standard of 72 classification tasks built on a carefully curated selection of datasets from the many thousands available on OpenML: the OpenML-CC18. It can be used as a drop-in replacement for many typical benchmarking setups. These datasets are deliberately medium-sized for practical reasons. An overview of the benchmark suite can be found at `https://www.openml.org/s/99` and the Appendix. We first describe the design criteria of the OpenML-CC18 before discussing uses of the benchmark and success stories.[1,6]

## 6.1 Design Criteria

The OpenML-CC18 contains all verified and publicly licenced OpenML datasets until mid-2018 that satisfy a large set of clear requirements for thorough yet practical benchmarking:

(a) The number of observations is between $500$ and $100\,000$ to focus on medium-sized datasets that can be used to train models on almost any computing hardware.

(b) The dataset has less than 5000 features, counted after one-hot-encoding categorical features (which is the most frequent way to deal with categorical variables), to avoid most memory issues.

(c) The target attribute has at least two classes, with no class of less than 20 observations. This ensures sufficient samples per class per fold when running 10-fold cross-validation experiments.

(d) The ratio of the minority and majority class is above $0.05$ (to eliminate highly imbalanced datasets which require special treatment for both algorithms and evaluation measures).

(e) The dataset is not sparse because not all machine learning models can handle them gracefully, this constraint facilitates our goal of wide applicability.

(f) The dataset does not require taking time dependency between samples into account, e.g., time series or data streams, as this is often not implemented in standard machine learning libraries. As a precaution, we also removed datasets where each sample constitutes a single data stream.

(g) The dataset does not require grouped sampling. Such datasets would contain multiple data points for one subject and require that all data points for a subject are put into the same data split for evaluation. We introduce this constraint and the one above to simplify usage of the datasets, as one does not have to use specialized cross-validation procedures.

We also applied several more opinionated criteria to avoid issues with problematic datasets:

---

[5]Development is carried out on GitHub. Contributor guidance can be found at `https://docs.openml.org`.

[6]The OpenML-CC18 is the successor of a preliminary benchmarking study called OpenML100, containing 100 classification datasets, and fixes several issues we encountered when working with the OpenML100.

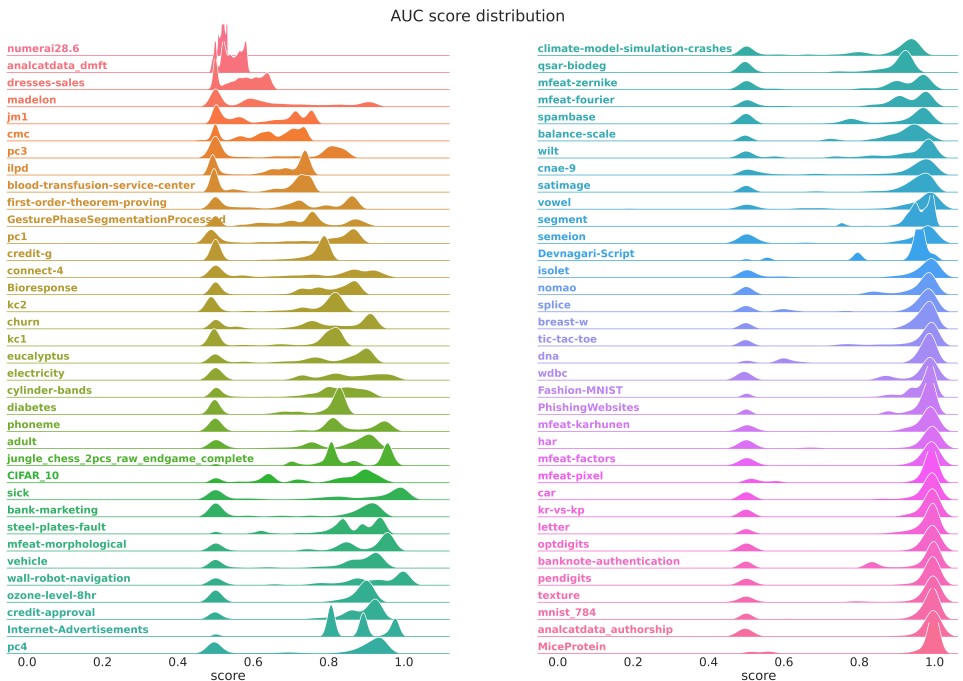

Figure 3: Distribution of the scores (average area under ROC curve, weighted by class support) of 3.8 million experiments with thousands of machine learning pipelines, shared on the CC18 benchmark tasks. Some tasks prove harder than others, some have wide score ranges, and for all there exist models that perform poorly (0.5 AUC). Code to reproduce this figure (for any metric) is available on GitHub.[4]

(a) We strived to remove artificial datasets, as it is hard to reliably assess their difficulty. Admittedly, there is no perfect distinction between artificial and simulated datasets (for example, a lot of phenomena can be simulated that can be as simple as an artificial dataset). Therefore, we removed datasets if we were in doubt of whether they are simulated or artificial.

(b) We removed datasets which are a subset of larger datasets. Allowing subsets would be very subjective, as there is no objective choice of a dataset subset size or a subset of the variables or classes.

(c) We excluded tasks for which the original target feature has been transformed or changed, e.g., when classes of a categorical target feature were merged or when a continuous target feature (for original regression tasks) was discretized to create a classification task.

(d) We removed datasets without any source or reference. We want to be able to learn more about their intended use and how to interpret learned models, and avoid black box datasets.

Finally, to ensure that datasets are sufficiently challenging, we applied the following restrictions:

(a) We removed datasets which can be perfectly classified by a single attribute or a decision stump, as they do not allow us to meaningfully compare machine learning algorithms.

(b) We removed datasets where a decision tree could achieve 100% accuracy on a 10-fold cross-validation task, to remove datasets which can be solved by a simple algorithm which is prone to overfitting training data. We found that this is a good indicator of too easy datasets. Obviously, other datasets will appear easy for several algorithms, and we aim to learn more about the characteristics of such datasets in future studies.

We created the OpenML-CC18 as a first, practical benchmark suite. In hindsight, we acknowledge that our initial selection still contains several mistakes. Concretely, *sick* is a newer version of the *hypothyroid* dataset with several classes merged, *electricity* has time-related features, *balance_scale* is an artificial dataset and *mnist_784* requires grouping samples by writers. We will correct these mistakes in new versions of this suite and also screen the more than 900 new datasets that were uploaded to OpenML since the creation of the OpenML-CC18. Moreover, to avoid the risk of

overfitting on a specific benchmark, and to include feedback from the community, we plan to create a dynamic benchmark with regular release updates that evolve with the machine learning field. We want to clarify that while we include some datasets which may have ethical concerns, we do not expect this to have an impact if the suite is used responsibly (i.e., the benchmark suite is used for its intended purpose of benchmarking algorithms, and not to construct models to be used in real-world applications).

## 6.2  Usage of the OpenML-CC18

The OpenML-CC18 has been acknowledged and used in various studies.[1] For instance, Van Wolputte and Blockeel [2020] used it to study iterative imputation algorithms for imputing missing values, König et al. [2020] used it to develop methods to improve upon uncertainty quantification of machine learning classifiers and De Bie et al. [2020] introduced deep networks for learning meta-features, which they computed for all OpenML-CC18 datasets. In some cases, the authors needed a filtered subset of the OpenML-CC18, which is natively supported in most OpenML clients. Other uses of the OpenML-CC18 include interpreting its multiclass datasets as multi-arm contextual bandit problems [Bibaut et al., 2021a,b] and using the individual columns to test quantile sketch algorithms [Mitchell et al., 2021].

Cardoso et al. [2021] claim that the machine learning community has a strong focus on algorithmic development, and advocate a more data-centric approach. To this end, they studied the OpenML-CC18 utilizing methods from Item Response Theory to determine which datasets are hard for many classifiers. After analyzing 60 of its datasets (excluding the largest), they find that the OpenML-CC18 consists of both easy and hard datasets. They conclude that the suite is not very challenging as a whole, but that it includes many appropriate datasets to distinguish good classifiers from bad classifiers, and then propose two subsets: one that can be considered challenging, and one subset to replicate the behavior of the full suite. The careful analysis and subsequent proposed updates are a nice example of the natural evolution of benchmarking suites.

For completeness, we also briefly mention uses of OpenML100, a predecessor of the OpenML-CC18 that includes 100 datasets and less strict constraints. Fabra-Boluda et al. [2020] use this suite to build a taxonomy of classifiers. They argue that the taxonomies provided by the community can be misleading, and therefore learn taxonomies to cluster classifiers based on predictive behavior. van Rijn and Hutter [2018] and Probst et al. [2019a] used it to quantify the hyperparameter importance of machine learning algorithms, while Probst et al. [2019b] used it to learn the best strategy for tuning random forest based on large-scale experiments (although Probst et al. [2019a] and Probst et al. [2019b] use only the binary datasets without missing values).

Based upon these works, we conclude that the OpenML-CC18 is being used to facilitate very diverse directions of machine learning research.

## 7  Further OpenML Benchmarking Suites

We now review other OpenML benchmarking suites. For this, we focus on AutoML benchmarking suites, but also provide examples of others.

## 7.1  The AutoML Benchmark Suite

The AutoML benchmark [Gijsbers et al., 2019] also makes use of an OpenML benchmark suite to evaluate AutoML tools in a reproducible manner. Combined with code to automatically run experiments, any of the integrated AutoML tools can be evaluated on any suitable OpenML task or suite directly from the command line.

### 7.1.1  Benchmark Suite Design

The AutoML benchmark explicitly sources part of their datasets from the OpenML-CC18, but also includes datasets used in AutoML competitions (primarily Guyon et al. [2019]) or previous comparisons of AutoML systems. A step-by-step list of recreating the benchmark suite does not exist, but general guidelines are provided. Since the original release in 2019, the AutoML benchmark has

been extending their selection of datasets.[7,8] In the discussion below, aspects which are specific to the newer selection are indicated with an asterisk (*).

The suite shares some of its design criteria with OpenML-CC18, such as the minimum number of instances, as well as the exclusion of artificial datasets and those which require grouped sampling. However, it loosens some other restrictions specifically because of the assumption that AutoML tools should be able to deal with additional complexities:

(a) There is no limit to 100 000 instances or 5000 features, tools can restrict themselves to learners which scale well or use, e.g., low-fidelity estimates.
(b) There is no limit for class imbalance, tools can use their preferred techniques to deal with imbalanced data (e.g., SMOTE [Chawla et al., 2002]).
(c) It includes sparse data, though it is currently converted to dense format for tools that don't support sparse data.*
(d) It includes a suite of regression problems.*

Some other restrictions are instead stricter because of the tabular AutoML context:

(a) The "easy dataset" filter also takes into account results from OpenML across various learners, to try to avoid datasets which need little search beyond algorithm selection.
(b) The number of image classification problems is explicitly restricted, as they are typically better solved with Deep Learning and the benchmark's focus is tabular AutoML tools.

Similar to OpenML-CC18, the AutoML benchmark suite is intended to be regularly updated to reflect modern day challenges and to avoid overfitting.

### 7.1.2 Usage of the AutoML Benchmark Suite

Before the introduction of the AutoML benchmark suite, the closest to an accepted standard for tabular AutoML benchmarking was the set of datasets on which Auto-WEKA was originally evaluated [Thornton et al., 2013]. This selection of tasks was still used in, e.g., Mohr et al. [2018] and consisted of 21 problems, a third of which are image classification tasks which are typically not the intended use-case for the AutoML tools. However, it was by no means a standard. For example, Drori et al. [2018], Rakotoarison et al. [2019] and Gil et al. [2018], all published at the same workshop, each used different selections of datasets.

The original AutoML benchmark suite has been used in multiple AutoML publications, either as is [LeDell and Poirier, 2020, Wang et al., 2021, Feurer et al., 2021a] or with modifications. Sometimes more datasets are used, as Zöller and Huber [2021] combine it with OpenML-CC18 and OpenML100 and Kadra et al. [2021] add datasets from UCI and Kaggle. For the latter, hold-out evaluation is used instead of the suite-defined 10-fold cross-validation. Erickson et al. [2020] use additional datasets from Kaggle competitions to compare directly to solutions proposed by human competitors.

Other times not all datasets in the benchmark suite are used, e.g., Zimmer et al. [2021] uses all but four big datasets for computational reasons, while Parmentier et al. [2019] limit themselves to only four of the big datasets in the suite to assess their method designed for big datasets. Mohr and Wever [2021] omitted some datasets because of technical issues.

### 7.2 Further Existing OpenML Benchmarking Suites

OpenML contains other benchmark suites as well, such as the OpenML100-friendly that only contains the subset of the OpenML100 without missing values and with only numerical features, or Foreign Exchange data for machine learning research [Schut et al., 2019].

We invite the community to create additional benchmarks suites for other tasks besides classification, for larger datasets or more high-dimensional ones, for imbalanced or extremely noisy datasets, as well as for text, time series, and many other types of data. We are confident that benchmarking suites will help standardize evaluation and track progress in many subfields of machine learning, and also intend to create new suites and make it ever easier for others to do so.

---

[7] Announcement of the new suites: `https://github.com/openml/automlbenchmark/issues/187`

[8] `https://www.openml.org/s/{218,269,271}` are the original, regression, and expanded suite, respectively

# 8 Limitations and Future Work

As benchmarking suites are increasingly being picked up by the machine learning community, we also observed several limitations that should be tackled in future work.

**Overfitting.** While it has not yet been demonstrated, we assume that as more methods are being evaluated on benchmarking suites, overfitting on fixed suites is increasingly likely. We therefore aim to periodically update existing suites with new datasets that follow the specifications laid out by the benchmark designers (e.g., as done for computer vision research [Recht et al., 2019]) and invite the community to extend existing suites with harder tasks, as done in NLP research [Kiela et al., 2021].

**Credit Assignment.** Curating a benchmark is a lot of work, and we have manually inspected and corrected datasets for the OpenML-CC18 over the course of multiple months. It is therefore important to give proper credit to everyone involved in creating benchmarking suites, for example by somehow making benchmarking suites citable.

**Automating the curation of useful suites.** We are not aware of any related work that describes how to curate machine learning benchmark suites. In this paper we have defined benchmarking suites by formalizing objective, but also more subjective constraints. Providing automated ways to create high quality, diverse and realistic benchmarking suites is thus an important, open research question. Additionally, post-hoc research, such as the one conducted by Cardoso et al. [2021], is important to check the validity of benchmarking suites, and we hope for more such techniques to be developed and also to become applicable during the suite design process.

**Computational issues.** While studying applications of the OpenML-CC18 in Section 6.2 we realized that even though we consciously focused on mid-size datasets, some larger ones still incurred too high computational load, so some researchers have used subsets of the OpenML-CC18 in their work. Future suites could more carefully trade off the completeness of benchmarking suites and computational issues, for example by choosing representative subsets [Cardoso et al., 2021].

**Breadth of current benchmarking suites.** On the other hand, many researchers are interested in benchmarking larger (deep learning) models on larger datasets from many domains (including language and vision). We are working on ways to enable the creation of such benchmarking suites as well, and welcome further involvement from the community.

**Specification of resource constraints.** The task and suite specifications do not yet allow for constraints on resources, e.g., memory or time limits. Specific benchmark studies could impose identical hardware requirements, e.g., to compare running times. Where requiring identical hardware is impractical, general constraints would ensure results are more comparable when multiple people run their experiments on a suite. Explicit constraints also help interpret earlier results.

**Disclosure of ethical issues** We currently encourage creators to disclose any ethical concerns with datasets in their benchmark suite in its description. In the future we want to support this natively on a dataset level (e.g., by integrating datasheets [Gebru et al., 2018]) and benchmark suite level (by providing a dedicated information field).

# 9 Conclusion

Our goal is to simplify the creation of well-designed benchmarks to push machine learning research forward. More than just creating and sharing benchmarks, we want to allow anyone to effortlessly run and publish their own benchmarking results and organize them online in a single place where they can be easily explored, downloaded, shared, compared, and analyzed. We created a new benchmarking layer on the OpenML platform that allows scientists to do all the above with just a few lines of code. We then introduced the OpenML-CC18, a benchmark suite created with these tools for general classification benchmarking.

The use of suites is further motivated by a closer look at the AutoML benchmark suite. We also reviewed how other scientists have adopted these benchmarking suites in their own work, from which it becomes clear that a continuous conversation with the research community is essential to evolve benchmarks and make them better and more useful over time. We hope that this work will unleash a rapid evolution of benchmarks suites and large-scale studies that teach us more about machine learning than any single study could.

**Acknowledgements** This work has partly been funded by the German Federal Ministry of Education and Research (BMBF) under grant no. 01IS18036A, by the Deutsche Forschungsgemeinschaft (DFG, German Research Foundation) – 460135501 (NFDI project MaRDI), by the European Research Council (ERC) under the European Union's Horizon 2020 research and innovation programme under grant no. 716721 (Beyond BlackBox) and 952215 (TAILOR), through grant #2015/03986-0 from the São Paulo Research Foundation (FAPESP), by AFRL and DARPA under contract FA8750-17-C-0141, as well as through the Priority Programme Autonomous Learning (SPP 1527, grant HU 1900/3-1) and Collaborative Research Center SFB 876/A3 from the German Research Foundation (DFG).

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
