# OpenReview forum: "OpenML Benchmarking Suites"
_NeurIPS.cc/2021/Track/Datasets_and_Benchmarks/Round2 — NeurIPS 2021 Datasets and Benchmarks Track (Round 2)_

### Official Review · Reviewer_BcsL · 2021-09-18
**This work is mainly about introducing and using the OpenML benchmarking suites, so I think the contributions are not sufficient for acceptance.**

**Rating:** 4
**Confidence:** 3
**Clarity:** Yes, the paper is well written.

**Strengths:**

1. This article presents the OpenML software tools that help to create and leverage machine learning benchmarking suites.
2. Clear explanations and illustrations about OpenML and OpenML benchmarking suites.

**Weaknesses:**

1. The main contribution of this article is introducing the software tools which help to create and leverage OpenML benchmarking suites. It's not suitable for this conference topic.
2. The article covers a lot of the introduction of OpenML benchmarking suites, but not much of the main work on machine learning.

**Additional Feedback:**

The authors are recommended to provide more technical details in the supplementary materials.

**Correctness:**

This article uses two benchmarking suites and their applications to illustrate their approach. They don't provide  evaluation methods or experiment design.

**Documentation:**

The authors give sufficient detail and the URL to access the code and benchmarking suite.

**Ethics:**

There are no ethical concerns for the submission.

**Relation To Prior Work:**

No, this work doesn't compare with previous contributions.

**Summary And Contributions:**

OpenML is a collaborative platform , and OpenML benchmarking suites are a set of OpenML tasks which  makes experiments performed on them comparable and reproducible. This article advocates the use of curated, comprehensive suites of ML datasets to standardize benchmarking, and provides software tools to easily create and use these benchmarking suites. The software tools simplify the creation of well-designed benchmarks to push machine learning research forward.

---

> ### Author Response · Authors · 2021-09-25
> **Response to reviewer BcsL**
>
> Thank you for your review, we will address the points in the order they were raised.
>
>
> With all due respect, we think that our submission does in fact exactly fit the description of the call for papers of the Datasets and Benchmarks Track, for each major contribution:
>  - the code contribution of the new benchmark suite feature to OpenML and the tools for (semi-)automatically creating curated suites would fall under ‘benchmark tools’. The CFP reads: "Data generators, reinforcement learning environments, or benchmarking tools are also in scope".
>  - the contribution of a concrete benchmark suite to use for machine learning and the discussion of how it was created (OpenML-CC18 was not previously published at a peer reviewed venue) falls under the following from the CFP: “In addition to new datasets and benchmarks on new or existing datasets, we welcome submissions that detail advanced practices in data collection and curation that are of general interest”.
>
> When we look at already accepted papers from the first round we can see accepted papers with either type of contribution (https://blog.neurips.cc/2021/08/12/updates-on-the-neurips-2021-datasets-and-benchmarks-track/), for example Li et al. (https://openreview.net/forum?id=9E3dTIMxL8S), Otness et al. ( https://openreview.net/forum?id=pY9MHwmrymR) and Liang et al. ( https://openreview.net/forum?id=izzQAL8BciY) present benchmark suites and Narayan et al. (https://openreview.net/forum?id=hwjnu6qW7E4) and Banbury et al. ( https://openreview.net/forum?id=8RxxwAut1BI) provide benchmark tools.
>
> We would appreciate it if you could be more concrete about which specific details you missed and would like to see in the supplementary materials.

---

### Official Review · Reviewer_e37p · 2021-09-19
**Solid idea, but not very interesting**

**Rating:** 5
**Confidence:** 4
**Correctness:** The claims seem correct.
**Clarity:** The paper is very clearly written.

**Strengths:**

This seems like a good idea in that it makes it even easier to share datasets on which an experiment was run. In the past, one might have had to create lists of datasets and tasks and share them in a custom format.

**Weaknesses:**

This is a very good idea and a great "feature" to add to OpenML. It will undoubtedly save time. However, the conceptual novelty is limited.

**Additional Feedback:**

Keep up the great work at OpenML -- it is very valuable and appreciated!

**Documentation:**

The documentation seems adequate, though from the main OpenML website it was not obvious how to browse the set of suites.

**Relation To Prior Work:**

Prior work is discussed.

**Summary And Contributions:**

The popular AutoML repository makes it easy for researchers to find and experiment on data in a reproducible manner. This paper discusses the addition of the the notion of benchmark suites to the AutoML dataset repository. A benchmark suite is a set of datasets and tasks on those datasets that can then be used for further reproducibility across multiple datasets, without having to share the lists of tasks, etc.

---

> ### Author Response · Authors · 2021-09-25
> **Response to reviewer e37p**
>
>
> Thank you for your review, we will address the points in the order they were raised.
>
> We are happy to hear that you find that this is a good idea, that it helps standardized benchmarking and saves time. Regarding conceptual novelty, we believe this is twofold. First, we propose a new practical and scalable approach that allows anyone to very easily create standardized and curated benchmark suites. That is, benchmarks that match specific sets of technical and scientific constraints, which hasn’t been done consistently before to the best of our knowledge. With this, we hope to create a new sense of critical thinking about standardized benchmarking, and that by providing the tools and platform to easily create, share, compare, and use curated benchmarking suites, particularly well-designed suites will gain adoption in many scientific communities. With standardized benchmarking also comes a more standardized analysis, which makes interpreting empirical research easier, and will hopefully accelerate real progress. While dataset repositories existed before, to the best of our knowledge, being able to easily design them using clear a-priori constraints, and to couple them with evaluation strategies (including the data splits which are indispensable for reproducibility) in a machine readable and extensible way is a new contribution. This was only possible by building on OpenML as a platform to host the datasets and suites, as well as novel code contributions, such as the notebooks/tools that help with (semi-)automated benchmark suite curation. We will also extend the paper with a more clear process description of how these tools can be used for new benchmark suites.
>
> Second, we present concrete benchmark suites, e.g. the OpenML-CC18 suite which we have carefully curated, as a widely applicable ML benchmark suite. CC18 has not been published at a peer-reviewed venue, and we will also further clarify this in the paper.
>
> We will update new.openml.org to make the benchmark suites easier to find, thank you for the suggestion.

---

### Official Review · Reviewer_q36C · 2021-09-20
**Useful platform for organizing benchmarking studies**

**Rating:** 8
**Confidence:** 5

**Strengths:**

In my opinion, the OpenML platform has large potential to push the boundaries of organizing and curating benchmark studies for the domain ML learning since it has been around for some time and contains a large pool of already completed experimental results as well as descriptors of datasets and methods used to produce results. Given this, it has the potential to support standardization of benchmarking for ML. The platform itself is very accessible and easy to use and contribute to.

**Weaknesses:**

In order to become a leader in benchmarking for the domain of ML, the platform will need to be extended in terms of coverage of representation of different types of ML tasks. The authors can spend some space in the paper to discuss how easy or complicated this is given the large number of ML tasks that researchers are considering in their work. The number of tasks currently covered is somehow limited. Another important aspect that is only covered a bit in the discussion is a clarification of how curation is currently performed in the definition of ML benchmarking in OpenML. Do any defined and documented curation protocols (or workflows) exist so far in OpenML? I believe defining such protocols will have a crucial role in the standardized benchmarking that OpenML is aiming for.

**Additional Feedback:**

- Please clarify and define the terminology you are using at the beginning of the paper and use it consistently.
- Please clarify your understanding of a task.
- Please clarify are there any curation protocols currently that are used for defining benchmarking studies in OpenML.
- Please discuss how novel tasks can be included in OpenML in order to define benchmarking studies on top of those tasks.
- Please discuss in a sentence or two the choice of design criteria for the two example benchmarking studies you have presented. How were they selected?

**Clarity:**

The paper is relatively clear to follow. The authors should improve the introduction of the terminology at the beginning as it is not crystal clear what is meant under task, flow, run and experiment and should use the terms consistently throughout the paper. For example: at a point, you claimed that benchmarking suites are collections of tasks, then at another point you suits are collections of ML datasets.

**Correctness:**

All the claims made in the submission are correct. The authors presented two concrete benchmarks (one for classification and one for AutoML) and clearly described the design criteria behind the benchmarks as well as examples of usage of both benchmarks.

**Documentation:**

The OpenML platform is very well documented and the authors provide a lot of useful links in the paper and the supplementary material. Slightly more information needs to be explicitly provided on licensing and maintenance of the platform,

**Ethics:**

There are no ethical concerns with this paper.

**Relation To Prior Work:**

The paper clearly discusses the related work and gives a brief historical overview of benchmarking in ML as well as a short discussion of what is missing.

**Summary And Contributions:**

This paper presents both a platform for organizing ML benchmarks as well as two examples of specific benchmarks, one for classification and one for AutoML. The authors introduce a novel benchmarking layer on top of the OpenML platform, which is integrated with its APIs and allows the organization and creation of so-called benchmarking suits. The paper claims several contributions:  advocacy of the use of curated and comprehensive benchmarking suits with the goal of benchmarking standardization, providing SW for creating and using the suits, discussion of example suits and their impact on ML research.

---

> ### Author Response · Authors · 2021-09-25
> **Response to reviewer q36C**
>
> Thank you for your review, we will address the points in the order they were raised.
>
> We are indeed currently working on extending OpenML to new types of data (e.g. audio/visual/textual data) and novel task types (e.g. fairness, object detection), however we consider this outside of scope for this specific paper. That said, as soon as these new types of datasets and tasks are included, it will be immediately possible to create benchmark suites for them since (as part of our contributions in the current paper) we have implemented tools in OpenML for the creation of benchmarking suites that are compatible across different tasks and dataset types.
>
> While we don’t yet provide explicit protocols for everyone to follow to create benchmark suites, we do provide the tools that allow people to create their own protocol for benchmark suite construction. In the paper, we gave an example of such a protocol for OpenML-CC18.
> Further standardization comes from using the benchmark suites, because tasks can explicitly capture information on e.g. dataset splits and performance metrics to use.  Based on your feedback we will add a section on a generalized protocol for creating benchmarking suites and add it to the paper.
>
>
> We did our best to provide license information, it’s on the website (https://new.openml.org/terms) as well as in the paper (footnote 2, page 3). However, we do recognize that clear license information was not present in every Github repository related to this paper and we have updated those repositories. For maintenance information, we have the documentation pages (https://docs.openml.org/), which include details about the OpenML governance model, as well as individual pages for the connector packages (e.g. openml-python’s can be found here: https://github.com/openml/openml-python/blob/develop/CONTRIBUTING.md). Is there any other information you missed or does this adequately address your concerns?
>
>
> Thanks for pointing out our inconsistent use of certain terminology. We will clarify some definitions here, though we will also update the paper as follows:
>  - a task is a (machine-readable) definition of an evaluation procedure for a specific dataset. Concretely, it contains information on the task type (e.g. classification or regression), the target feature (in the case of supervised problems), the evaluation procedure (e.g. k-fold CV, hold-out) and the specific splits for that procedure. It can also be annotated with tags, which are user-provided meta-information.
> - a suite is a curated collection of tasks with a description
> - a study is a collection of runs with a description
>
> To address your other concerns:
>  - as discussed above, we provide tools to create a curation protocol, and we give an example of an existing one when we introduce the OpenML-CC18
>  - new tasks can be created on any dataset (it is possible to upload new datasets in either ARFF or Parquet format), either through the web interface (`+’ in the top-right corner when you are logged in, though the forms still need to be updated in the new website) or any connector package (e.g. openml-python). If you are asking about new types of tasks (e.g. fairness tasks), then these need to be discussed because they require changes to the code base (e.g. the client APIs need to know how to read the task and new metrics may have to be implemented). To suggest a new type of task, you can raise an issue in the openml repository (there is a dedicated task type suggestion label). We will draw attention to this by adding instructions on the task type page on new.openml.org.
> - To construct OpenML-CC18 we opted for a practical set of guidelines that allow evaluations to be performed on a wide range of algorithms on a plethora of machines. Motivation for each decision is given in the bullet lists in the paper. If there are any specific design criteria that you find lack motivation, we would like to hear which, and hopefully we can clarify our choices.

---

### Official Review · Reviewer_a59r · 2021-09-20
**A new feature to group datasets in OpenML**

**Rating:** 6
**Confidence:** 5
**Correctness:** Yes

**Strengths:**

The paper is clearly written despite a few grammar issues. The proposed feature is useful and open source.

**Weaknesses:**

This paper describes an improved functionality, but very little above and beyond what is already possible with OpenML and/or its existing benchmark sets.

- it is not clear how a benchmark suite differs from a "Study" in current OpenML, or how this functionality improves existing ways of collating datasets in OpenML, for example by using tags.

- It is not clear how or why one might curate a benchmark suite from the currently available datasets on OpenML that escapes the problems mentioned here, e.g., cherry picking of datasets for specific studies to make an algorithm look good. Part of the advantage of a diverse set of benchmarks is the ability to characterize strengths AND weakness of algorithms, as the authors mention. And, a minimally filtered yet diverse set of benchmarks from OpenML have been published (OpenML100, OpenML-CC18), as the authors describe here. I'm left to conclude that benchmark suites may be useful for new collections of datasets. Yet this paper does not provide new datasets or suites, just a new way to group existing ones. Furthermore, the groupings given as examples have already been in use for a few years. The suite functionality may be nicer, but perhaps would be more appropriate as a release note or blog post.

- I believe pages 5 - 9 simply describe two existing benchmark suites that have already been published. The authors spend a lot of time rehashing this existing work. If these curations have not been published, the authors should make that clear.

- the paper ignores ethical concerns re: the potential for harm that could arise from benchmark suites curated to specific application areas, and / or those with privacy concerns.

**Additional Feedback:**

- the autogenerated figures on the openml website under Analysis could use work. The dots and font size are typically too small and the aspect ratio is typically too wide.

At the time of this writing, the analysis tab of OpenML-CC18 https://new.openml.org/search?type=study&study_type=run&id=99&sort=runs_included does not load for me in Firefox or Chrome.



**Clarity:**

Yes. Some notes:

- "too easy datasets" - "datasets that are too easy"

- May sentences beginning with "this", where the context for "this" is unclear

**Documentation:**

Yes

**Ethics:**

Yes. Ethics are ignored in the submission because the authors consider this "core research in the ML field". To me this isn't an excuse to ignore ethical implications of their work. There are clearly ethical issues to consider when curating collections of datasets, for example, how the curated set of datasets relate to and / or support real-world applications, and what those applications are. One could foresee the use of suites to curate datasets for potentially harmful applications in areas like algorithmic sentencing, lending and admissions. Indeed, as a platform for benchmark suites, it would be nice to see foresight on the part of the authors by incorporating meta-information on suites that  discuss ethical implications and/or intended uses.

**Relation To Prior Work:**

Yes, although it isn't totally clear how much easier the suite function makes benchmarking compared to what was used before

**Summary And Contributions:**

This paper proposes a "benchmark suites" feature addition to OpenML that allows users to group datasets into specific sets for curation. Certainly the suites feature (or something like it) is necessary and helpful, given that many datasets of varying degrees of quality are available and it is difficult to sift through them all. However I can't help but feel this contribution is quite limited.

---

> ### Author Response · Authors · 2021-09-25
> **Response to reviewer a59r**
>
> Thank you for your review, we will address the points in the order in which they were raised.
>
> The studies (and suites) feature and associated toolsets, as well as the OpenML100 and OpenML-CC18 suites, have not yet been published at a peer reviewed venue and are only described on arXiv; for that reason, we consider these a new contribution to the research community. We apologize that this was unclear in our submission and will update the paper accordingly. The AutoML benchmark has been published before, but we only present it as an example for a specific field and discuss its usage to further motivate the usefulness of benchmark suites and their practicality. We will clarify this in the paper.
>
> It sounds like you are familiar with earlier versions of our software (specifically using tags). The old style collections (with tags) are discontinued, since they do not allow a controlled creation of systematic benchmarks. The new benchmark suite and study implementations provide a permanent url with a dedicated description page, as well as clear and protected authorship: the user that created the suite is visible, and no other user may modify the study/suite, whereas anyone could tag new runs/tasks.
>
> More importantly, beyond simply collating datasets, with this paper we provide tools to create a systematic, curated benchmark. These make it easier to construct a suite based on explicit pre-designed criteria instead of considering datasets on an individual basis. This makes it easier to avoid (unintentional) cherry-picking. Granted, in the end it’s just a tool and there is no way to automatically bar malign actors from constructing a benchmark that is designed for their model. We hope to create a new sense of critical thinking about standardized benchmarking, and that by providing the tools and platform to easily create, share, compare, and use curated benchmarking suites, particularly well designed suites will gain adoption in their respective communities. When a clear standard for benchmarking suites is present in a community, it in turn becomes harder for malign actors to present a dishonest evaluation because a convincing motivation would need to be provided to deviate from the accepted standards.
>
> Thank you for raising the issue of ethical concerns. To address your issue, we will add an additional paragraph to the benchmark suites’ descriptions pointing out which datasets should be handled with care. In future work we plan to address this more natively by e.g. supporting datasheets and fairness based tasks, and adding a clear notice to datasets with known ethical concerns. In general we don’t believe the ability to construct benchmark suites using datasets which may have ethical issues is inherently a problem, as long as the *use* of these datasets is ethical. E.g. benchmarking to compare algorithm performance is fine, but building specific applications on dubious/unethical datasets is not.
>
> If you would be so kind as to provide specific line numbers where the use of the word “this” leads to confusion, we will try to rephrase our text to avoid that confusion. With only general directions it becomes hard to know which parts to revise exactly.
>
> Finally, the URL you provided is not valid as it tries to analyze runs on a task suite (which has no runs). The correct URL should be: https://new.openml.org/search?type=study&study_type=task&id=99
> If you could tell us how you reached the invalid URL it would be helpful for us to make sure users don’t accidentally access it in the future. We will also add a clear message on the website for such invalid URLs.

---

> > ### Comment · Reviewer_a59r · 2021-09-29
> > **Response to response**
> >
> > Thanks for your response. Whereas the responses in general look ok, I had expected an updated manuscript to be available for review before the end of discussion, that addressed the issues discussed above. Nevertheless, I still am left with the impression that the contribution of this paper is fairly small, especially for this venue.
> >
> > Responding to your comments:
> >
> > The clarification against tags makes sense. It should be mentioned in the paper that something "suite-like" was already available in OpenML, but was lacking permanence and provenance.
> >
> > I agree with the authors about the need for systematic benchmarks with explicit criteria. And, I do see why benchmark suites are useful. When evaluated from the point of view of novelty and significance of the contribution, however, I still find it to be too small to meet the stringent criteria of this conference and track. Part of the reason it lacks significance and novelty is the fact that dataset suites were already successfully curated without this contribution - as evidenced by the review of the suites that are already used in several papers.
> >
> > Ethical concerns: your response would be a satisfactory update to the paper.
> >
> > Lines with ambiguous or over-use of "This" :
> > - 23, 27, 39, 58, 228, 322
> >
> > Other misc. comments:
> > - Fig 3: "Code to reproduce this (for any metric) is available on GitHub." - the footnote is not on the bottom of the page, and footnote 3 (on page 5) is a link to openml docs.
> >
> >
> > Thoughts on improving the contribution in future manuscript versions:
> >
> > - Extend the analysis in Fig 3, and show that the curated suite(s) fix specific problems in the benchmarking ML of algorithms - for example, better characterization of the space of ML behaviors from different families of methods
> > - Define a set of suites that are tailored to analysis of specific problem domains/tasks or assessment of specific algorithms (beyond classification/regression)
> > - Clearer demonstration of how the suites functionality significantly improves benchmarking standards

---

> > > ### Author Response · Authors · 2021-09-30
> > > **Clarification**
> > >
> > > Thank you kindly for the response. We will update the manuscript shortly. We need to clarify  one point: “Part of the reason it lacks significance and novelty is the fact that dataset suites were already successfully curated without this contribution - as evidenced by the review of the suites that are already used in several papers”. This is not factually correct. In fact, the work that is presented here is *exactly* the work that enabled the creation of benchmarking suites, we simply never published it earlier.
> > >
> > > To clarify that point, here the full story:
> > > We created the initial benchmark suite functionality and the OpenML-CC18 suite about 3 years ago, but did not publish it in a peer reviewed venue at that time. Instead, we uploaded our work on arxiv [1] with the aim to advertise our early work and find active users of the OpenML-CC18 suite and the benchmark suite functionality (which at that time was in a very early development stage). The increased popularity and use of the suite helped us to gather user feedback, fix bugs and continuously improve OpenML's benchmarking suite functionality and the benchmarking suites themselves. Recently, having processed all user feedback - open source development can be time consuming - we made a stable release of the OpenML benchmarking suite functionality making it usable by a large audience. We are now pursuing a proper peer-reviewed publication for our work, including our carefully curated suite still called "OpenML-CC18". The NeurIPS Datasets and Benchmarking track provided an ideal venue for this work, hence the timing and the late submission of our work to this venue.
> > >
> > > Note that the existing arXiv article that introduced the OpenML-CC18 [1] should not be considered as a proper peer reviewed publication. To pass peer review, we needed evidence of the actual utility of the software functionality and the OpenML-CC18 to the research community. This takes time and in our case it took about 3 years since we introduced our very first OpenML-CC18 suite. By releasing our early work on arXiv, researchers outside the OpenML community now know about the benchmarking suites and are using them.
> > >
> > > We still believe that this work, which we have done and continuously improved in the past 3 years, deserves a proper peer reviewed publication. Since the concept is now sufficiently tested, and its utility has been established (you appear to also know the suite), it is the right time to make it known to a wide audience and start a community discussion on moving to more standardized benchmarking on a large scale. We would love for the community to start curating benchmarking suites for many more tasks and goals. Without a proper peer reviewed publication, uptake of curated standardized benchmarking in everyday research will surely be much slower. We also want to allow scientists to cite a peer-reviewed version of this work rather than the older arXiv article [1].
> > >
> > > We kindly request the reviewer to reflect on the following questions: Is a paper submitted to this venue automatically not novel if there is an earlier arXiv version that is already being cited and its content being used? Would it have been better not to introduce the OpenML-CC18 at an early stage in [1] and withhold/hide the benchmarking suite functionality on OpenML, just to make it suitable for publication now? We assume that the answer to both questions is no, and that there was only confusion about the timeline of this work, which we hope is now resolved.
> > >
> > > [1] https://arxiv.org/abs/1708.03731v2

---

> > > > ### Comment · Reviewer_a59r · 2021-10-01
> > > > **response to updated version**
> > > >
> > > > Thanks for the clarification and more about the history. The updated manuscript has addressed most of my concerns and I've updated my overall rating.
> > > >
> > > > I had assumed earlier work was done without the suites feature but have since reviewed the papers cited herein and the openml github. I can appreciate that a preprint should not count against novelty or contribution. However, I'm still concerned the 2017 preprint and subsequent publication of an updated version in 2021 is going to be incredibly confusing for others as well. As one example, the AutoML benchmark paper (ICML) cites the 2017 preprint as the work it is extending, and yet the current version of that same preprint (this paper) now cites and discusses the AutoML paper. I would encourage the authors to not create these confusing loops in the future. This paper could just as well be a unique manuscript that is upfront about reviewing the impact of the suites feature, explicitly claim the publishing novelty, and not be treated as an updated version of the 4 year old manuscript, which is now pretty deep in previous work.

---

### Author Response · Authors · 2021-09-29
**Updates and further feedback**

Dear reviewers,

Thank you again very much for your time reviewing our paper and the valuable feedback. We have now updated the website as promised:

1. It is now much easier to find benchmark suites and studies since we added them as a new category in the main menu panel on new.openml.org.
2. We added headers that briefly explain what suites/studies are, with a link to more documentation and how to create new ones.
3. We investigated the broken website reported by reviewer a59r and implemented a bugfix.

Moreover, we would like to make sure that we answered all open questions to your utmost satisfaction. As the discussion period will close at the end of the day, we would appreciate feedback on whether our responses have addressed the issues you raised. This would allow us to provide further details and clarifications before the discussion period closes. Otherwise, we hope that our responses convince you to update your scores.

Yours sincerely,
the authors

---

### Author Response · Authors · 2021-09-30
**Revised PDF available**

Dear reviewers,

We have uploaded a revised version of the manuscript, taking into account your feedback.

The most important change is to more clearly state the contributions of this work. These do not only include the benchmarking suite feature in OpenML, but also the OpenML-CC18 suite, which we’ve been developing over the past years and has not been in a peer-reviewed publication before. It should therefore be taken into account when judging the novelty and significance of this work. Other changes include e.g. clarifying terminology, fixing the broken footnote, addressing ethical concerns and updates to improve readability.
If your raised concerns have not yet been addressed in the revised version, it is likely because we simply were not able to make the changes yet (we will of course add everything promised in our responses).

Based on these updates, we kindly ask you to consider again whether the updates and responses warrant an update to the paper’s score. Nevertheless, please reach out if there are any remaining issues and questions.

Yours sincerely,
the authors

---

### Decision · Program_Chairs · 2021-10-09

**Decision:**

Accept

**Comment:**

This paper's review have quite high variance (4/5/6/8).

The reviewer who scored a 4 did not express high confidence (3) and their main concern with the paper is that it is out of scope as it details a software platform. I agree with the authors here that this paper is well within the scope of the track and am heavily downweighting this reviewer's score.

The reviewer who scored a 5 gave a very lightweight review with the main concern being limited conceptual novelty. I have read the authors response to this point, and believe it clarifies and addresses the point about novelty. The reviewer has clarified that they remain unconvinced that the paper is an interesting contribution, but that there are no major flaws that would block publication.

The two positive reviews (6/8) have provided more substantial constructive feedback for the authors most of which have been addressed.

After conferring with another AC I have decided to recommend acceptance of this paper.